# Communication Aware UAV Swarm Surveillance Based on Hierarchical Architecture

Chengtao Xu, Kai Zhang ⬤, Yushan Jiang, Shuteng Niu, Thomas Yang ⬤ and Houbing Song *⬤

Department of Electrical Engineering and Computer Science, Embry-Riddle Aeronautical University, Daytona Beach, FL 32114, USA; xuc3@my.erau.edu (C.X.); zhangk3@my.erau.edu (K.Z.); jiangy2@my.erau.edu (Y.J.); shutengn@my.erau.edu (S.N.); yang482@erau.edu (T.Y.)
* Correspondence: songh4@erau.edu

**Abstract:** Multi-agent unmanned aerial vehicle (UAV) teaming becomes an essential part in science mission, modern warfare surveillance, and disaster rescuing. This paper proposes a decentralized UAV swarm persistent monitoring strategy in realizing continuous sensing coverage and network service. A two-layer (high altitude and low altitude) UAV teaming hierarchical structure is adopted in realizing the accurate object tracking in the area of interest (AOI). By introducing the UAV communication channel model in its path planning, both centralized and decentralized control schemes would be evaluated in the waypoint tracking simulation. The UAV swarm network service and object tracking are measured by metrics of communication link quality and waypoints tracking accuracy. UAV swarm network connectivity are evaluated over different aspects, such as stability and volatility. The comparison of proposed algorithms is presented with simulations. The result shows that the decentralized scheme outperforms the centralized scheme in the mission of persistent surveillance, especially on maintaining the stability of inner UAV swarm network while tracking moving objects.

**Keywords:** persistent surveillance; hierarchical architecture of UAV teaming; communication aware UAV formation; dynamical object tracking; intercommunication quality

## 1. Introduction

Emerging new technologies on unmanned aerial vehicles (UAVs), such as 24/7 persistent surveillance [1], high-resolution sensing [2], autonomous UAV powered by machine learning [3,4], and secure UAV communication [5–8], have led to the changes of industries, academia, and government in investigating the strategies of rescuing, monitoring, and emergency communication construction. Persistent surveillance of UAVs gives the possibility of realizing continuous sensing coverage in an area of interest (AOI). High-resolution sensing on a different class of UAVs provides various levels of detail in the surveillance field. However, due to the multiple purposes of continuous monitoring, it is challenging for a single UAV agent to complete sensing tasks in a large-scale AOI. Therefore, the cooperative strategy using multiple UAVs is proposed to enhance the efficiency of continuous AOI surveillance. Meanwhile, in emergency scenarios such as wildfires and earthquakes, a minor delay of information sharing between multiple UAVs indicates more redundancy for agencies to process real-time tasks.

Swarm refers to a concept in which a group of coordinated, intelligent systems or agents are designed to achieve the same goal in a complex environment. It is not a simple accumulation of agents but a more redundant and efficient framework to achieve a high quality of service. UAV swarm has become a prevalent field of research in realizing persistent surveillance. Unlike research in the coordination between swarm agents in group path planning and barrier avoidance, AOI's continuous coverage with the UAV swarm faces various challenges in changing environments. For example, (a) the efficient surveillance task division within sensing group [9];

(b) optimization of the consuming energy of UAV [10–12]; (c) adjustment of surveillance plan based on the occasional natural disaster which has high priority [13].

To realize persistent surveillance with coordinated UAVs, three strategies have been considered in recent years: (i) on-duty UAVs replacement scheme with different recharging station distributions [14]; (ii) energy-efficiency path planning [15–18]; (iii) novel structures of UAV teaming [19]. UAV distribution's hierarchical architecture helps plan the efficient and adaptive surveillance while the mission is affected by the changing monitoring map which is caused by the weather condition or unseen factors. UAVs teaming's hierarchical structure intends to divide the surveillance task into different platforms such as ground station, high altitude UAV, and UAV sensing swarm. Each platform is in charge of different functions such as control, motion coordination, data transferring, and data package routing.

Figure 1 describes the hierarchical structure of the UAV surveillance team. Each platform marks its capabilities based on its difference in computation resources, communication system characteristics, and energy supply. High altitude UAVs equipped with a high-resolution camera are usually applied in the field sensing and the data exchange with ground stations since it has a relatively stable power supply compared to low altitude UAVs. However, due to its high altitude operation zone, the ground device with low transmission power makes it difficult to access its network service. To realize the economical and efficient operation of rescue missions and provide network service, a buffer layer of UAV swarm between high altitude UAV and ground objects extends this system's ability on connection service and real-time tracking. With many agents within the UAV swarm layer, an adaptive formation policy could be developed to fit in various requests. In [20], a distributed UAV surveillance scheme is applied in monitoring an AOI in terms of the event location. The covered area of each agent changes with the location of events. A ground station with a stable power supply is used as the terminal of data exchange. It obtains sensing and network service data from UAVs swarm or high altitude UAV in a specific design. A distance that exceeds the UAV swarm's communication range is considered impractical with a lower transmission power of low altitude UAV.

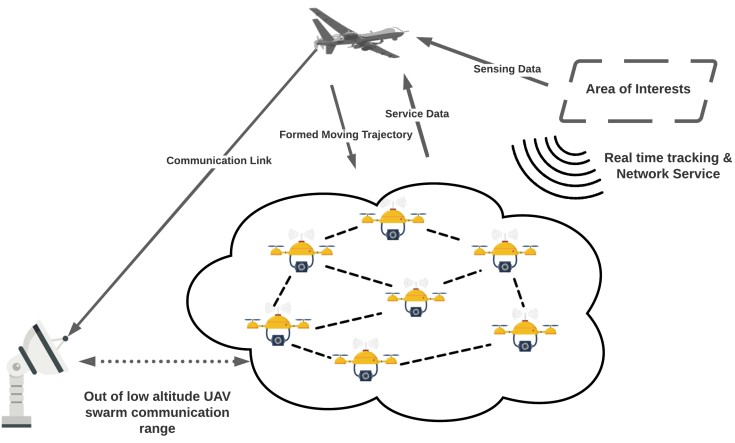

**Figure 1.** Hierarchical structure of UAV swarm based sensing and monitoring.

In [21], Li discussed a distributed coordination scheme of UAV swarm based on flock forming framework proposed by Reynold [22] in 1987, who described the basic flocking model consisting of three simple steering behaviors, separation, alignment, and cohesion. To realize the flock of UAV swarm, centralized control policies have been proposed [23–25] to realize implicit leader-follower teaming of UAVs. In [26], the authors consider a team distribution of leader-follower formation with a centralized control policy, in which the leader UAV collects the relative position of other following agents. The simulation results show

its effectiveness on block avoidance with continuous surveillance function. Feedback formation control scheme provides another avenue in trajectory formation with consensus velocity and thrust [27–29].

In persistent surveillance provided by the proposed UAV teaming structure, long-term monitoring and emergency communication and survivor tracking are the two application scenarios we considered in reality. Long-term monitoring requires the cooperation between low-altitude UAV swarm and high altitude platforms. To maximize the coverage area, the attraction values of the map cells are usually updated randomly [30–32]. The swarm searches on the existing map to find the shortest route to the target points. Meanwhile, the general attraction rate of each map cell changes regarding the swarm flying trajectory. Low altitude swarm extends the data-gathering capability of high-altitude UAVs by following a specific flying trajectory. The flying trajectories usually do not have priorities compared to the surveillance data quality under conditions.

However, the spreading patterns of natural disasters such as wildfires, earthquakes, and mudslides should be considered in the path planning. For example, in a wildfire, the percentage of forest coverage in the AOI influences the fire spreading rate between neighboring trees, which then affects the spreading time and direction [33]. A residential area in the direction of fire spreading needs longer staying time of UAV swarm to provide the network service, especially considering this in the phase of general path planning. The fire transition rate is the parameter dependent on the wind speed and forest coverage rate, assuming the residential area is under the direction of wind. Therefore, the assigned flying trajectory from high altitude UAV should follow the direction of wind. In our communication aware UAV swarm path planning, we would investigate the link quality variance with such a change of the mission's path planning.

We propose a framework (Figure 2) based on the hierarchical UAVs teaming architecture to perform a navigation task via the trajectory design according to changing map and decentralized control of low altitude sensing UAV swarm. It achieves navigation and trajectory functions by changing maps and decentralized control scheme of low altitude UAV swarm. Each UAV agent updates the state of communication link connecting the neighboring nodes. The consensus of agents' kinetic variable is based on its communication link quality, which forms a decentralized control scheme not relying on the leading device's coordination. With the remote sensing data on changing maps, the high altitude UAV generates an optimized path based on the low altitude swarm state. The swarm agents track each assigned waypoint by correcting its geometry center point. Real-time object tracking and network service data would be collected and transmitted to the terminal ground station by platforms with a greater power rate. The distributed control scheme of low altitude UAV swarm suggests that no coordination delay between each waypoint tracking and the idle link between low and high altitude platforms could enhance the package delivery rate or throughput on two layers. The simulation results show the ability of fast convergence of the link quality between agents and disturbance tolerance while following the assigned waypoints.

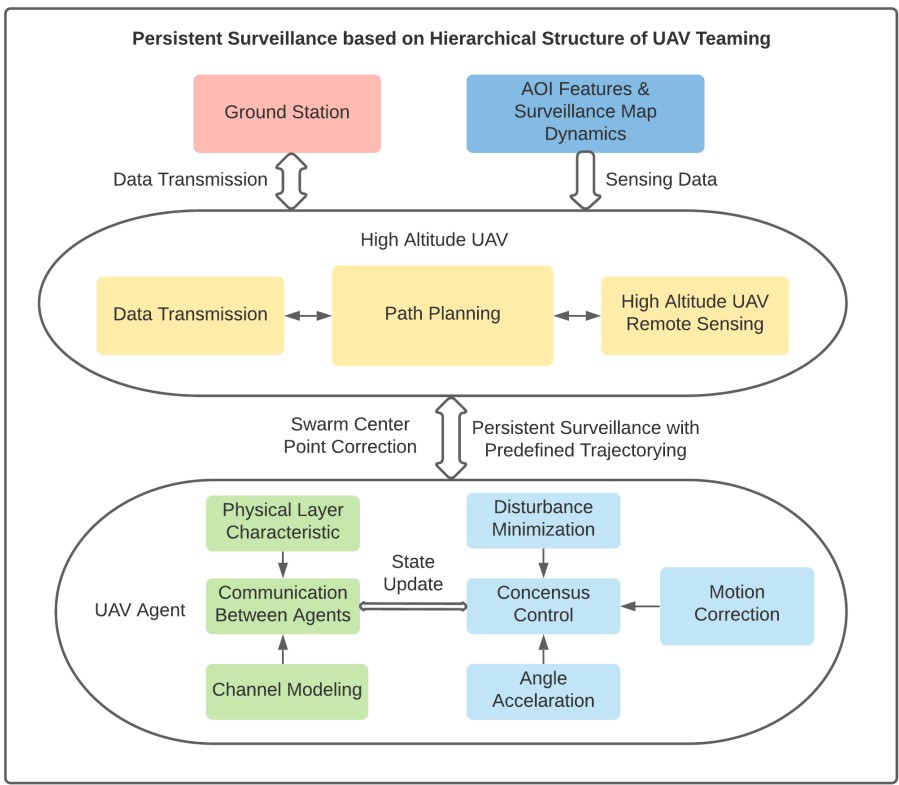

**Figure 2.** Function modules in each layer of platforms with hierarchical structure.

This paper studies the communication-aware formation of low altitude UAV swarm in applying persistent surveillance within hierarchical structure UAV teaming. Our schemes are proposed based on centralized and decentralized control policies considering the communication link quality between agents, and the waypoint tracking accuracy of the UAV swarm. The predefined trajectory for environment exploration would be discussed in response to different types of events. Exploring the desired area with different surveillance plans would result in various general real-time data transmission rates within the swarm network. The contributions of this work are summarized as follows:

(1) We compared the centralized and decentralized control scheme in studying the UAV swarm's waypoint tracking function while considering its inner communication link quality, which is a practical technical challenge needed to be solved to realize applicable UAV swarm-based persistent surveillance.

(2) We compared the performance of UAV swarm trajectory following algorithm based on the centralized and decentralized position updating strategy to achieve better waypoint tracking accuracy.

(3) We evaluated the robustness of swarm internal communication link, formation stability with the disturbance in chasing the wanted waypoint.

The rest of the paper focuses on the dynamics and communication of low altitude UAV swarm, and the contents are organized as follows. The channel state model of low altitude UAV agent and consensus control of low altitude UAV swarm are presented and discussed in detail in Section 2. Section 3 gives the simulation results of centralized and distributed control on waypoint tracking and disturbance tolerance of low latitude UAV swarm on following the trajectory with a changing sensing environment. Finally, we present the conclusions and present our future work in Section 4.

## 2. Problem Formulation

### 2.1. Communication Channel State for UAV Agent Formation Control

We assume that multiple inputs and multiple outputs (MIMO) antennas are adopted on low-altitude UAV agents to access the AOI ground users. As shown in Figure 3, the sensing and network service 'cloud' is formed by the UAV swarm nodes. Each sensing node exchanges the information with its neighboring nodes with one or multiple communication channels. This one or multiple communication channels forms the communication between agents. The quality of such a communication link could reveal the overall performance of swarm network service, which is also called quality of service (QoS). Our model neglects the cochannel interference on the signal receiver side, which is caused by the difference in transmission power.

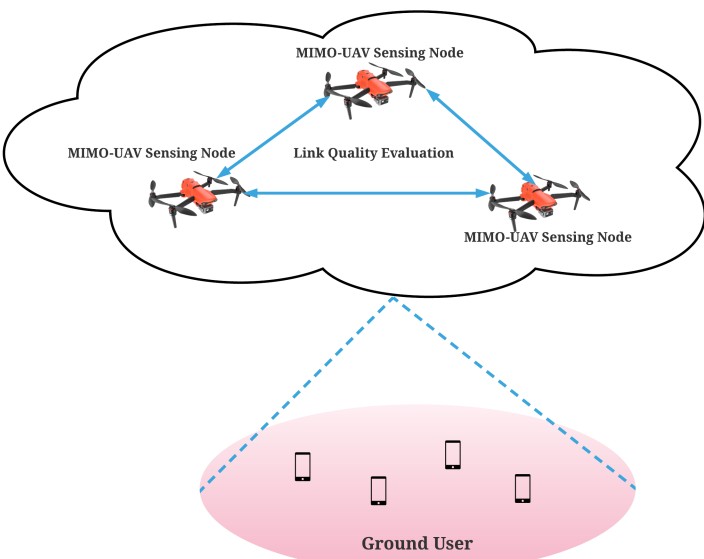

**Figure 3.** Low altitude UAV swarm communication and sensing network.

Moreover, we assume the adaptive filter on a single receiver's side could suppress the cochannel signal and environment noise. Therefore, to evaluate the communication channel quality between 2 UAV agents, we simplified the problem to purely measure the single input and single output (SISO) transceiver and receiver link. One UAV could maintain multiple SISO links with its neighboring UAV sensing nodes.

We denote the position of each UAV as $q_i$ in an multiagent system with $n$ agents, the dynamics of each UAV is given by $\dot{q}_i = u_i$, where $\dot{q}_i, u_i \in R^3$. It simplifies the single UAV dynamics, which considers the angle accelerations provided on yaw, roll, pitch. $u_i$ denotes the control of the $i$th UAV based on the current channel state with its neighboring agent. $r_{ij}$ indicates the distance between agent $i$ and $j$ for rigid formation [34], which represents the neighboring agents keep a prescribed desired distance with each other.

$$r_{ij} = \left\| q_i - q_j \right\|_2 \tag{1}$$

The reception probability of data package transmission is defined in Equation (2), a critical wireless channel parameter for a SISO communication link. It is used to evaluate the communication channel state between each neighboring UAV. As the SISO model shows in Figure 4, $\alpha$ denotes the antenna characteristics, which stays the same on the sides of transceiver and receiver. $\beta$ is the required data rate fed into the digital signal processor (DSP). $v$ is the path loss exponent related to the environment. $r_0$ is the distance of causing near field effects between neighboring antennas. The data package reception probability indicates the rate of successful communication between transceiver and receiver. Therefore,

to evaluate the SISO communication channel quality between *i*th and *j*th UAV, firstly we have the reception probability denoted by

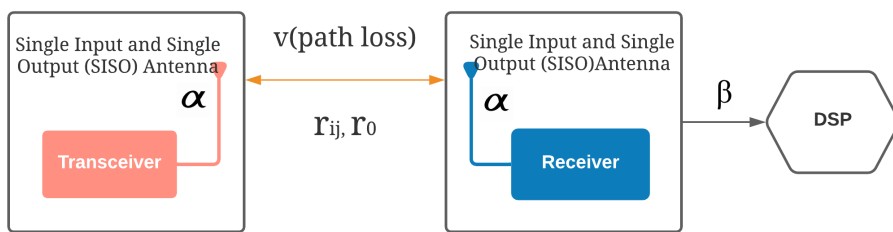

**Figure 4.** SISO communication channel model between UAV agents.

$$a_{ij} = exp(-\alpha(2^\beta - 1)(\frac{r_{ij}}{r_0})^v) \tag{2}$$

where $a_{ij}$, the package reception probability measuring the possibility that the receiver receiving sensing information accurately from transceiver. In the swarm's dynamics, it evaluates the probability of influence from transceiver to the receiver side.

In the domination of modeling SISO antenna, the far field is majorly considered with the path loss effect of the channel, which decreases the packages reception rate with the increase of radio propagation distance. However, in forming a UAV swarm, the near field effects happen more easily between agents. It refers to the severe mutual interference of SISO antenna that significantly degrades the communication performance of swarm communication network. A simplified approximation model used in describing the near fields effects is defined as:

$$g_{ij} = \frac{r_{ij}}{\sqrt{r_{ij}^2 + r_0^2}} \tag{3}$$

here, $r_{ij}$ denotes the distance between *i*th and *j*th agent, $r_0$ refers the distance of causing the near field effects between agents. When $r_{ij} \to 0$, $g_{ij}$ goes to 0, which characterizes the interference effect in the near field. When $r_{ij} \gg r_0$, $g_{ij}$ is nearly 1, which implies the near field effects can be ignored in the antenna far field. Then, communication performance indicator between neighboring UAVs can be defined as:

$$\phi_{ij} = -\beta\left(\left(vr_{ij}\right)^{v+2} + \left(\beta vr_{ij}^v + r_0^v\right)r_0^2\right)\frac{e^{-\beta}}{r_0^v}g_{ij} \tag{4}$$

To define the communication aware formation controller, the potential function is used in evaluating the interaction between neighboring agents. The artificial potential function is defined here for measuring the difference between performance indicator and its reference:

$$\psi(r_{ij}) = \phi^* - \phi_{ij} \tag{5}$$

where the function $\phi^*$ is the reference value of the optimized value of the communication performance indicator, then, the gradient of $\phi_{ij}$ is computed as:

$$\nabla_{q_i}\phi(r_{ij}) = P(a_{ij}) \cdot e_{ij} \tag{6}$$

where $e_{ij} = (q_i - q_j)/r_{ij}$, and $P$ is function of package acceptance rate for determine the neighboring nodes of agent *i*. Therefore, the gradient of velocity or control of UAV agent *i* can be written as:

$$u_i = \nabla_{q_i}V_i = \phi_{ij}\psi(r_{ij}) \tag{7}$$

The neighbor of the *i*th UAV is defined as those channels having a package acceptance rate larger than the threshold *th*. The receiver throws the transmitted package away when

the channel quality lower than *th* since the potential data lose with bad channel quality. So the function $P(a_{ij})$ can be defined as

$$P(a_{ij}) = \begin{cases} 1, & a_{ij} \geq th \\ 0, & a_{ij} < th \end{cases} \tag{8}$$

In [35], the author designed the formation controller based on the variance of communication performance indicator, in which each agent follows the control input's gradient direction to reach the lower value of $\phi_{ij}$.

### 2.2. Cooperative Control of UAV Swarm in Trajectory Following with Centralized Agent Speed Control

Unlike target point searching of a single UAV agent, the UAV swarm's group behavior is hard to control in adjusting the swarm center to reach the target point. Simultaneously, the UAV agent also has to maintain the communication link performance to satisfy the package reception rate condition on the receiver side. Here, we discuss centralized position updating methods for the control of UAV agent's speed.

#### 2.2.1. Target Point Searching

For low altitude UAV swarm shown in Figure 5, assuming one target point within the AOI is assigned by the high altitude platform as the surveillance task in a given time. Maintaining the distance between each UAV by controlling communication link quality, the swarm center is expected to get closed to the wanted mission point. Therefore, it is necessary to calculate the difference between swarm center and target point position as one input of the swarm control function. To calculate the center of swarm $C_t$ at time $t$, we have

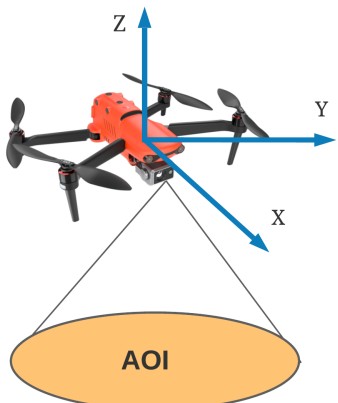

**Figure 5.** Low altitude UAV agent model.

$$C_t = \{x_t, y_t, z_t\} \tag{9}$$

$$x_t = \frac{\sum_{i=1}^{n} x_i}{n}, y_t = \frac{\sum_{i=1}^{n} y_i}{n}, z_t = \frac{\sum_{i=1}^{n} z_i}{n}, \tag{10}$$

where $n$ is the swarm size, $x_i$ is the position of $i$th low Altitude UAV on $x$ axis, $y_i$ is the position of $i$th low altitude UAV on $y$ axis, $z_i$ is the position of $i$th low altitude UAV on $z$ axis. The target point position is denoted as

$$T = \{x_T, y_T, z_T\} \tag{11}$$

Distance between center of swarm at time $t$ and target point can be denoted as:

$$r_{c_t} = \|C_t - T\|_2. \tag{12}$$

In order to optimize the search efficiency, an artificial potential function is designed to evaluate the distance between target point and the center of swarm.

$$\eta(r_{C_t}) = -\delta(r_{C_t}), \forall (i,j) \in n \tag{13}$$

The target point searching controller makes the group move in the direction of minimizing the difference between swarm center and target point.

$$\arg\min_{r_{c_t}} \chi(r_{c_t}) = \left\{ r_{c_t} \mid \chi(r_{c_t}) = \min_{r'_{c_t}} \chi(r'_{c_t}) \right\} \tag{14}$$

The gradient of $\delta(r_{C_t})$ is computed as:

$$\nabla_{C_t} \delta(r_{C_t}) = \frac{C_t - T}{r_{C_t}} \tag{15}$$

Therefore, the target searching control function can be defined as:

$$u_{C_t} = \nabla_{C_t} \delta(r_{C_t}) \tag{16}$$

Therefore, the control function of agent i at moment t could be written as

$$u_t = u_{C_t} + u_i \tag{17}$$

### 2.2.2. Waypoint Tracking

We assume multiple waypoints are on the assigned surveillance mission by high altitude UAV. The UAV swarm is assigned a time to reach the waypoint in AOI. With disruptions from environment and swarm agent position adjustment caused by maintaining the communication link's stability, the assigned waypoint reaching time could not be enough for the UAV swarm to move toward the new waypoint. Therefore, we propose an alternate centralized waypoint tracking scheme in the path planning of the UAV swarm.

We consider two waypoints, *k* and *l*, that are assigned consecutive waypoints. The corresponding positions are $T_k$ and $T_l$. The distance between center of swarm and node *k* and *l* can be denoted as:

$$\begin{aligned} r_{tk} &= \|C_t - T_k\|_2 \\ r_{tl} &= \|C_t - T_l\|_2. \end{aligned} \tag{18}$$

The artificial potential functions are designed to evaluate the distance between center point and two waypoints.

$$\begin{aligned} \kappa(r_{tk}) &= -\delta(r_{tk}) \\ \kappa(r_{tl}) &= -\delta(r_{tl}) \end{aligned} \tag{19}$$

To reach both two waypoints, we need

$$\arg\min_{r_{ik}} \chi(r_{tk}) = \left\{ r_{tk} \mid \chi(r_{tk}) = \min_{r'_{ik}} \chi(r'_{tk}) \right\} \tag{20}$$

$$\arg\min_{r_{ll}} \chi(r_{tl}) = \left\{ r_{tl} \mid \chi(r_{tl}) = \min_{r'_{il}} \chi(r'_{tl}) \right\}. \tag{21}$$

Considering the controller functions between center point to both waypoints are nearly same, we have

$$\nabla_{C_t} \delta(r_{tl}) \approx \nabla_{C_t} \delta(r_{tk}) = m_{c_t}, \tag{22}$$

where the $m_{C_t}$ is the average of two control inputs, it could be denoted as

$$m_{c_i} = \frac{2C_t - (T_k + T_l)}{r_{tk} + r_{tl}} u_{kl} = m_{c_i}. \tag{23}$$

The control of reaching the following waypoint is

$$u = u_{kl} + u_i. \tag{24}$$

The agent $i$'s control includes the speed updating component from maintaining the neighboring communication link quality and thrust of reaching waypoints.

*2.3. Cooperative Control of UAV Swarm in Trajectory Following with Decentralized Agent Speed Control*

Different from target points searching, a trajectory is formed by a set of continuous waypoints, which is hard to find the average value of waypoints to adjust the swarm center to locate on the path. Such a center position updating of the swarm is recognized as a centralized control scheme in realizing the waypoint tracking function, which is less effective for the path planning of a large-scale UAV swarm. Therefore, we let the position updating be made by every agent rather than leveraging an identical control setting over agents based on the difference between the swarm center and the predefined waypoint.

The distance from $i$th agent to waypoint at time $t$ is denoted as

$$r_i = \|q_i - f(t)\|_2, \tag{25}$$

in which $r_i, f(t) \in R^3$ represents the distance from agent $i$ to the waypoint of trajectory at time $t$ and trajectory formed by high altitude UAV in terms of surveillance needs.

The artificial potential function under this distributed formation control is

$$\varphi(r_i) = -\varsigma(r_i)\varsigma(r_i) = q_i - f(t). \tag{26}$$

where $f(t)$ is used as a reference value for agent $i$th position.

The control function with agent based position difference could be written as:

$$S_i = \frac{\varphi(r_i)}{r_i}. \tag{27}$$

Therefore, we have the control on the agent $i$ by following the waypoint on a trajectory:

$$u = S_i + u_i \tag{28}$$

## 3. Simulation Results and Analysis

This section simulates the model based on the proposed target point searching and waypoints tracking algorithms in MATLAB from a centralized and decentralized control scheme. Pseudo codes are presented to explain the simulation results. Evaluation metrics such as communication indicator value, the distance between agents, and distance from swarm center to the wanted position are defined and used on evaluating the performance of UAV swarm in the mission of persistent surveillance. In the simulation, we assume the weather condition is ideal for UAV operation, which means the wind flow does not affect the UAV dynamics. There are no projectiles, such as birds or barriers, in the environment, which could block the flying path of UAV. For UAV swarm's actual application in persistent surveillance, battery capacity is essential to accomplish the mission. Increasing the UAV agent's control gain will increase the device's power consumption, which lowers the efficiency of the UAV operation. Here, in the simulation, each UAV agent's battery capacity is large enough to satisfy the flight operation time.

To realize the low latency communication within the UAV swarm network, especially in providing emergency network service, a stable communication link between agents helps avoid the data package transmission congestion. Under the requirement of high information exchange rate in swarm networks, a sudden lousy communication quality with existing routing algorithms [36,37] could cause the uncontrollable package flooding

phenomenon. Therefore, we consider an accumulated index in evaluating the UAV swarm communication quality

$$\zeta_i(t) = \left\{ t \mid \sum_{j=1}^{n} \phi_{ij}, i \in n, j \in n \right\}.$$ (29)

in which, $\zeta$ is calculated as the sum of communication indicator in terms of all the channel links between $i$th UAV and its neighboring agents. A sudden bad channel state between $i$th UAV and one near UAV agent could cause the increment of indicator value. Therefore, the sum of the communication indicator's value could reveal QoS fluctuation with UAV swarm dynamics.

Of the near-field effects introduced in our SISO model for evaluating the communication link quality, a closer distance between UAV agents would give a more significant value of the communication indicator $\phi_{i,j}$, which suggests the significant portion of interference in the data exchanging between UAVs. In our model, the control portion of maintaining a stable communication link suppresses this interference. Therefore, the drone within the swarm could maintain the distance to prevent them colliding with each other. From the experiment results, the swarm can also maintain a stable distribution in the 3D space. Although it does not organize the team with a leader and follower format within the swarm surveillance group, to stable geometry distribution of swarm agent, we choose the distance from agent 1 to other UAVs to measure the movement's dynamical stability of the swarm. It is also observed from the experiment results that the control of maintaining the appropriate distance between UAV agents has higher priorities than reaching the wanted surveillance location.

$$r_{1j} = \left\| q_1 - q_j \right\| \quad j \in s.$$ (30)

The difference between swarm geometry center and waypoint expresses our method's tracking accuracy, in which this metric has the same expressing format as Formula (12).

With limited assigned waypoint transition time by higher altitude platforms, a trade-off between tracking and maintaining communication link stability is inevitable. However, it could be regulated by the value of the package acceptance rate threshold.

Therefore, three evaluation parameters are proposed to evaluate the performance of UAV swarm waypoint tracking algorithm. $r_{C_t}$ measures the accuracy of the swarm waypoint tracking function. Since agent one usually operates in the geometry center of 7 UAV swarm, $r_{1j}$ is adopted to reveal the swarm's stability while approaching the wanted waypoint. $\zeta_i(t)$, the sum of the communication indicator value through the seven UAV agents, is used on revealing the fluctuation of communication link while the tracking waypoint of the UAV swarm is changing.

The following simulation will use several UAV agents to experiment with proposed algorithms. Seven UAV agents are specially picked as the benchmark of swarm agent quantity setting. Compared to the number of agents lower than seven, it has relatively suitable complexity in describing UAV swarm dynamics. For example, especially on maintaining communication link performance, four UAV agents setting in the experiment cannot present the problem of centralized formation control with the increasing of swarm agent quantity. It is also not suitable for deploying UAV swarm in the ongoing surveillance mission required to provide high-speed network accessing service. In the end, the simulation of a decentralized control scheme would focus on the scalability aspects of UAV swarm in waypoint tracking.

### 3.1. Target Point Searching

The agents' speeds are considered to be adjusted with two components: maintaining communication link quality and gradually approaching the assigned target point. The process is expressed in Algorithm 1.

---

**Algorithm 1:** Target Point Searching

---

**Result:** Maintaining communication link while approaching the target point
initialization:
$T = (10, 10, 10)\%$ Assigning Target Points
$q_i =$ random value $< 100$
$speed_i = 0$
**while** *t < assigned surveillance time* **do**
  Sum of agent position $= \sum_{i=1}^{m} q_i$
  $C_t = \frac{\text{sum of agent position}}{\text{swarm size}}$
  Difference between center and target point $D_t = C_t - T$
  **while** *i < swarm size* **do**
    **while** *j < swarm size (j! = i)* **do**
      $r_{ij} =$ norm $(q_i - q_j)$
      $a_{ij} = \exp\left(-\alpha\left(2^\beta - 1\right)\left(\frac{r_{ij}}{r_0}\right)^v\right)$
      **if** $a_{ij} >= P$ **then**
        $\phi_{ij} = -\beta\left(\left(vr_{ij}\right)^{v+2} + \left(\beta vr_{ij}^v + r_0^v\right)r_0^2\right)\frac{e^{-\beta}}{r_0^v}g_{ij}$
      **else**
        $\phi_{ij} = 0$
      **end**
    **end**
    $e_{ij} = \left(q_i - q_j\right)/r_{ij}$
    $\nabla_{q_i}\phi\left(r_{ij}\right) = \phi\left(r_{ij}\right) \cdot e_{ij}$
    $u_c = \frac{C_t - T}{\|C_t - T\|_2}$
    $u_i = u_c + \nabla_{q_i}\phi\left(r_{ij}\right)$ % calculate the control of each agent
    $speed_i = speed_i + u_i$
    $q_i = q_i + speed_i * step\_size$
  **end**
**end**

---

In each time step, the UAV's control would be adjusted based on the communication indicator $\phi_{ij}$, which depends on the package acceptance rate threshold $P$. The lower the package acceptance rate $P$ holds, the greater the control value to maintain the team format on symmetrical distribution. An initial randomized swarm agent position did not show the constraints of the proposed algorithm in the following simulation.

In Figure 6, each line of color represents a moving UAV trajectory, which gradually gets close to the target point. It starts with a randomized position in the space. In the early phase, due to the enormous control brought by the difference between the swarm center and the target point, each agent's velocity update concentrates on getting close to the target. While the group center is getting close to the wanted position, the trajectory with curve shows the phase of UAV revising its position to build up a better communication link. When both criteria become stable, the final result shows that all seven agents could be evenly distributed on the space with (10, 10, 10) as their center point. Meanwhile, as shown in Figure 7, agent 7's (the centering agent) trajectory shows the adjustment of UAV swarm could be brought by the mismatch of swarm center and target point.

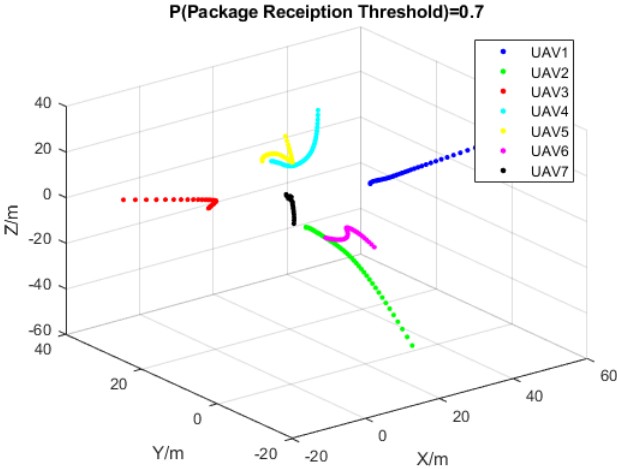

**Figure 6.** UAV swarm searching path to target point (10, 10, 10).

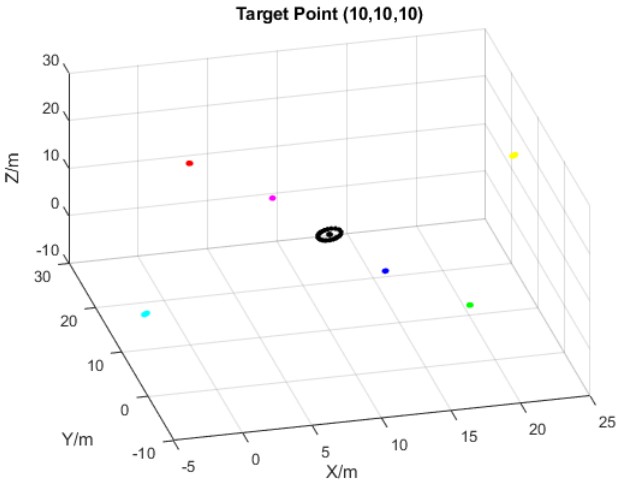

**Figure 7.** Position maintaining of swarm on the top of surveillance target.

Besides, based on the simulation result, we found out that $P = 0.7$ could be the optimal value of the application rate threshold in this algorithm, which gives the optimal control factor to revise the position of UAV to achieve the convergence of communication indicator value. In Figure 8, the fast convergence of $\phi_{ij}$ is based on such an threshold value.

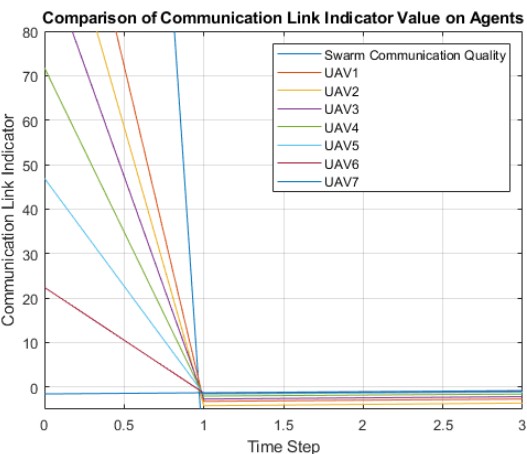

**Figure 8.** Convergent communication indicator of each UAV agent.

### 3.2. Waypoint Tracking Based on Centralized Control Scheme

In the early phase of the UAV swarm persistent surveillance, after reaching the initial target point, to monitor the AOI, the low altitude UAV swarm is assigned to go over waypoints predetermined by the high altitude UAV platform. Here, we assume the waypoints are a set of points evenly distributed on the surveillance area, which requires the UAV swarm to reach the point in a given amount of time decided by the path planning platform. High altitude UAV gives those waypoints with its remote sensing data based on the multi types of sensors attached to it in real applications. UAV swarm assigned to specific waypoints is desired to realize the function of real-time tracking of an object, providing network service. Generally, a transition time of moving the swarm from one waypoint to the other depends on the task priorities and distance from the current waypoint [38–40]. Therefore, the transition time between waypoints is an essential aspect of evaluating the waypoint tracking performance. With insufficient transition time, the policy of UAV swarm tracking should be revised as well.

Figure 9 shows the trajectory of UAV swarm agent. Swarm center and waypoint are updating with the proposed target searching function in waypoint tracking. The red line represents the trajectory of the swarm center at each simulation time step. Waypoint distribution is designed as a hexagon shape to verify the swarm's ability to expand its surveillance region in AOI. As expected, the swarm center trajectory would pass every waypoint and form the same shape as a hexagon. However, the UAV agent's trajectory and the swarm center do not present the ability to track those waypoints with high accuracy.

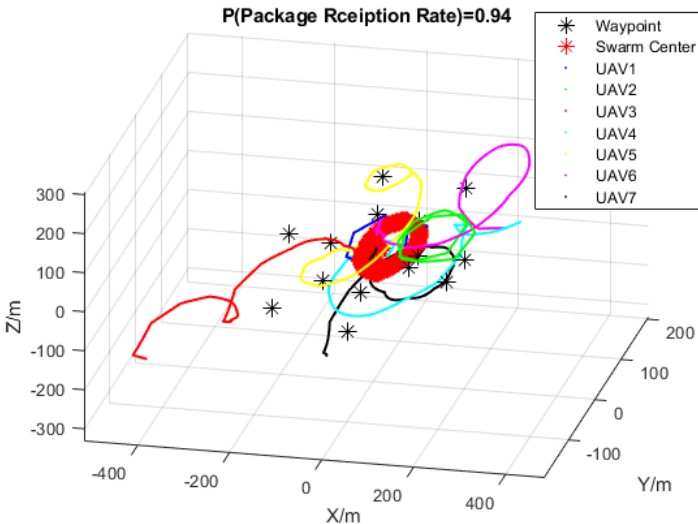

**Figure 9.** Swarm trajectory, center point, waypoint updating with centralized position updating policy.

In Figure 10, the distance between agent one and others varies in the form of erratic curves. The distance between UAV1 and UAV3, UAV1 and UAV7 increases with adding new waypoints in the surveillance area. To monitor AOI's wider area, the distribution of waypoints could only be more sparse in our simulation. As time goes by, the distance between UAV3, UAV7, and the remaining swarm agent cannot support the swarm in forming an effective surveillance network. Eventually, the lost control UAV3 and UAV7 will lose the connection with the surveillance group. In Figure 11, the swarm center to waypoint, yet, show a decreasing of the distance difference on time step smaller than 200. The outer hexagon layer of the waypoint reduces the tracking accuracy of the UAV swarm after 200-time steps.

The target searching algorithm is less effective in realizing the waypoint tracking of UAV in the early phase of surveillance. Therefore, we proposed the following Algorithm 2 to solve swarm losing control in the transition of waypoints. The alternative waypoint tracking algorithm is based on the centralized control scheme. It compresses the update portion of the communication indicator value because the algorithm block limitation is in one page.

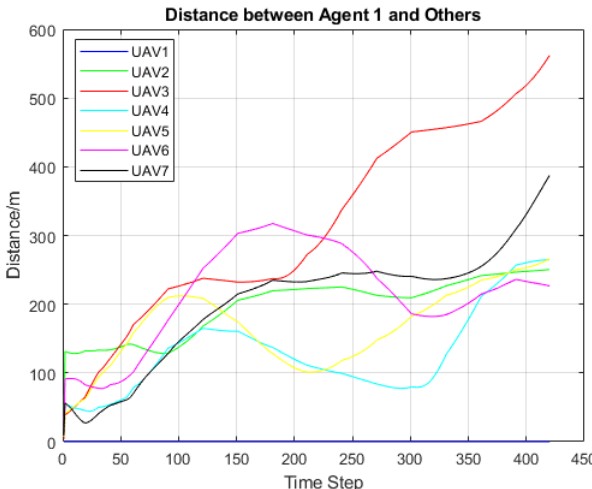

**Figure 10.** Distance between Agent 1 and others with centralized control policy.

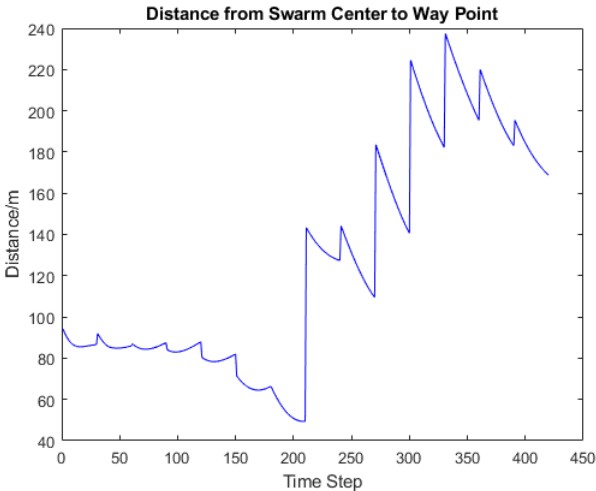

**Figure 11.** Distance from swarm Center to Waypoint with centralized control policy.

In Algorithm 2, before moving forward from one waypoint to the others, $\varepsilon$ is used to evaluate whether the next waypoint is reachable for the swarm agent. Otherwise, a replaced waypoint closer to the swarm's current position would be applied in adjusting the speed of each UAV.

In Figure 12, even the swarm shows similar tracking accuracy, the UAV agent trajectory shows the convergence of distance between UAV agents. A more reliable channel link between UAV agents could be built with a distance from 20 to 50 m, which prevents the collision between agents. The distance varies between agents when tracking the near waypoint. It could be seen from Figure 13. Besides the swarm center adjustment, the communication aware control intends to pull its neighboring agent together in the process of waypoint transition, making this alternated position control scheme reliable. The alternative waypoint makes the control priorities become maintaining the stable distribution of the UAV swarm. In Figure 14, instead of presenting a divergence trend shown in Figure 11, while the waypoint moves from the inner hexagon circle to the outer one, the new control scheme suppresses the divergence trend and keeps their distance difference under 160 m. The communication link quality is still not promised, as shown in Figure 15.

---

**Algorithm 2:** Waypoint Tracking

---

**Result:** Maintaining communication link while transmitting between waypoints.

initialization:

$q_i = (x_i, y_i, z_i)$% Randomized Position Value

$speed_i = 0$

**while** *number of investigated waypoint < number of planned waypoints* **do**

    Sum of agent position $= \sum_{i=1}^{m} q_i$

    $C_t = \frac{\text{sum of agent position}}{\text{swarm size}}$

    $T(k+1)$ = Assigned waypoint position at time k + 1

    $T(k)$ = Assigned waypoint position at time k

    Difference between center and current waypoint $T(k)$: $D_t = \|C_t - T(k)\|_2$

    % In tracking next waypoint

    **if** $D_t < \varepsilon$ **then**

        **while** *t < assigned surveillance time* **do**

            $e_{ij} = (q_i - q_j)/r_{ij}$

            $\nabla_{q_i} \phi(r_{ij}) = \phi(r_{ij}) \cdot e_{ij}$

            $u_c = \frac{C_t - T(k+1)}{\|C_t - T(k+1)\|_2}$

            $u_i = u_c + \nabla_{q_i} \phi(r_{ij})$ % calculate the control of each agent

            $speed_i = speed_i + u_i$

            $q_i = q_i + speed_i * step\_size$

        **end**

    **else**

        % Compute middle point of straight line between current waypoint
and next waypoint

        $T_{avg} = \frac{T_k + T_{k+1}}{2}$

        $e_{ij} = (q_i - q_j)/r_{ij}$

        $\nabla_{q_i} \phi(r_{ij}) = \phi(r_{ij}) \cdot e_{ij}$

        $u_c = \frac{C_t - T_{avg}}{\|C_t - T_{avg}\|_2}$

        $u_i = u_c + \nabla_{q_i} \phi(r_{ij})$ % calculate the control of each agent

        $speed_i = speed_i + u_i$

        $q_i = q_i + speed_i * step\_size$

    **end**

**end**

---

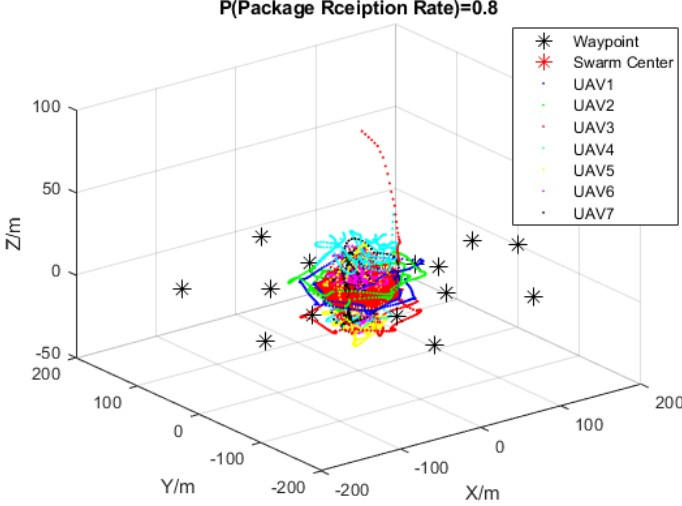

**Figure 12.** Swarm dynamics with an alternated centralized control algorithm.

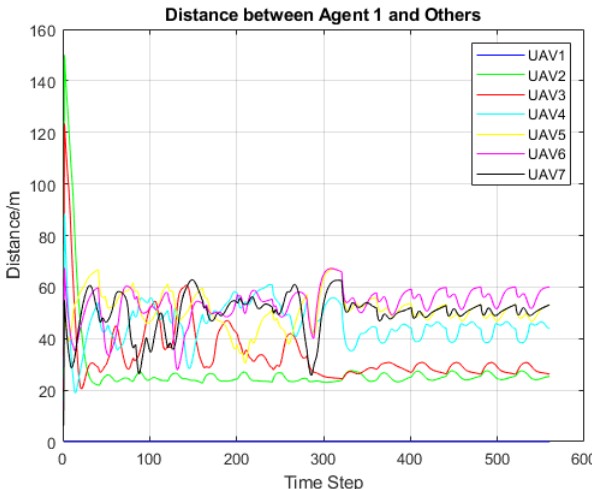

**Figure 13.** Distance between agent 1 and Others with alternative centralized policy.

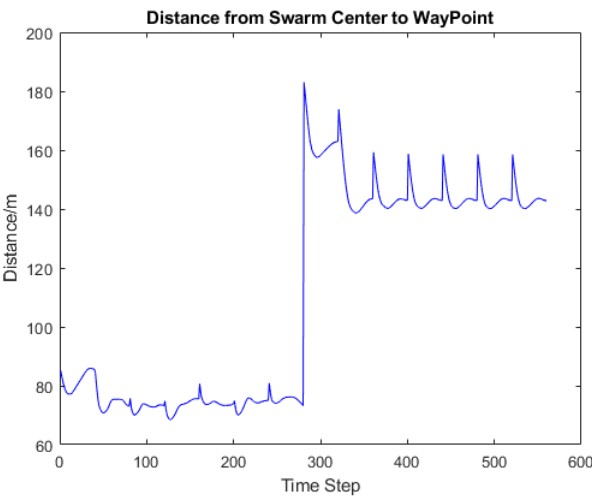

**Figure 14.** Distance between Swarm Center and Waypoint with alternated centralized policy.

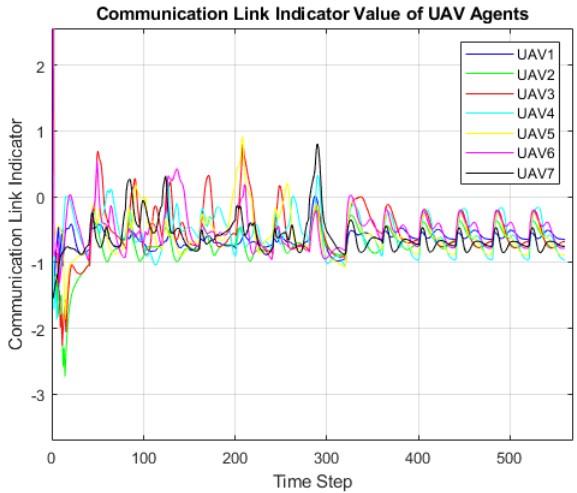

**Figure 15.** Communication link indicator variance with alternative centralizing waypoint tracking algorithm.

### 3.3. Decentralized Control Scheme of UAV Swarm in Waypoint Tracking

An assigned trajectory is combined with dense waypoints, requiring UAV agent to have accurate control on adjusting its position to maintain the communication quality

and waypoint tracking accuracy in surveillance. The centralized swarm center position comparing the algorithm above is not effective in working in such a task. Therefore, instead of giving each agent the same control inputs based on the swarm center position comparing result, the decentralized control policy compares the UAV agent position with the assigned waypoint in the trajectory at each time step. The algorithm is explained as follows.

---

**Algorithm 3:** Decentralized Control Scheme on Waypoint Tracking

---

**Result:** Maintaining communication link while following the assigned waypoint
initialization:

$q_i = (x_i, y_i, z_i)$% Randomized Position Value
$speed_i = 0$
**while** *Past Surveillance Point < Assigned Surveillance Point* **do**

   **while** *t < assigned surveillance time* **do**

      $T = f(t)$
      Sum of agent position = $\sum_{i=1}^{m} q_i$
      Difference between agent position and target point $D_t = q_i - T$
      **while** *i < swarm size* **do**

         **while** *j < swarm size (j! = i)* **do**

            $r_{ij} = \text{norm}\,(q_i - q_j)$
            $a_{ij} = \exp\left(-\alpha\left(2^\beta - 1\right)\left(\frac{r_{ij}}{r_0}\right)^v\right)$
            **if** $a_{ij} >= P$ **then**
               $\phi_{ij} = -\beta\left(\left(vr_{ij}\right)^{v+2} + \left(\beta vr_{ij}^v + r_0^v\right)r_0^2\right)\frac{e^{-\beta}}{r_0^v}g_{ij}$
            **else**
               $\phi_{ij} = 0$
            **end**
            $e_{ij} = (q_i - q_j)/r_{ij}$
            $\nabla_{q_i}\phi(r_{ij}) = \phi(r_{ij}) \cdot e_{ij}$
            $speed_i = speed_i + \nabla_{q_i}\phi(r_{ij})$
         **end**
         $u_c = \frac{D_t}{\|D_t\|_2}$
         $u_i = u_c + \nabla_{q_i}\phi(r_{ij})$ % calculate the control of each agent
         $speed_i = speed_i + u_i$
         $q_i = q_i + speed_i * step\_size$
      **end**
   **end**
**end**

---

Each UAV agent updates its control portion without relying on consensus control introduced by the difference between UAV swarm center and waypoint position. It neglects to collect each agent's position information in the swarm and calculates the control input needed for each one of them, which helps solve the problem brought by the information exchange delay and instability due to the loss of control command data. Meanwhile, the computation efficiency for UAV forming a surveillance team is improved by assigning the position comparing process on each independent platform within the swarm. The control portion relies on two aspects, the agent's position difference to waypoint and the communication link quality. The designed algorithm maintains each agent's communication link quality.

Meanwhile, link quality stability promises the distance between agents is also stable and will not collide. In other words, the mechanism makes the agent group stay steady on the top of the wanted waypoint. When the waypoint update is assigned, the position comparing mechanism will move the whole group to the new surveillance area again. The following simulation results would explain these effects.

Figures 16–18 show the moving trajectory of seven UAV agents and the swarm center with each time step. Each UAV agent owns the unique dynamical behavior and moves forward

with keeping the distance to maintain the communication link's stability. The swarm center could perfectly follow the straight-line trajectory predefined by the high altitude UAV platform. Simulation result from Figure 18 shows swarm adjusts its center after each waypoint updating. With enough transition time given between each waypoint, the distance difference could be lower than 10 m. Most of the time, the center position difference keeps between 20 to 30 m, which is caused by not enough assigned transition and surveillance time. It could be seen from Figure 17 that the UAV agent keeps the distance between each at the range from 20 m to 50 m, which could be an appropriate length for an amateur drone to hold the communication link with its neighboring agent. For example, the furthest range of existing amateur UAV platforms (e.g., RyzeTech Tello Quadcopter, SNAPTAIN A10 MiniFoldable Drone) is 300 m to 400 m. The range of causing near field interference of SISO antenna was assumed to be 2 m based on the relation between radio wavelength and antenna diameter. Therefore, a communication range between UAV agents from 20 m to 50 m gives a promising distance for high data rate communication. Furthermore, the distance variance between the agent one and others stays in the range of 10 m.

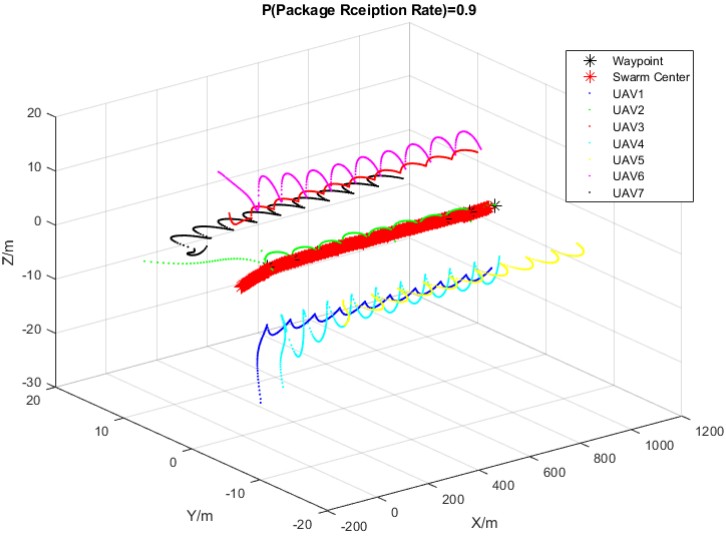

**Figure 16.** Swarm trajectory, center point updating with straight-line trajectory under decentralized control policy.

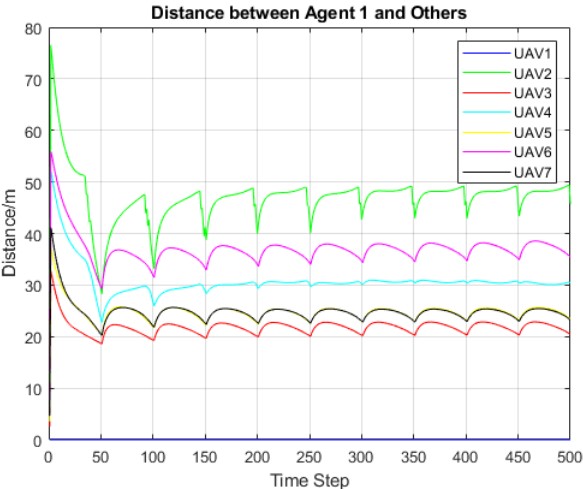

**Figure 17.** Distance between agent 1 and others with decentralized control policy.

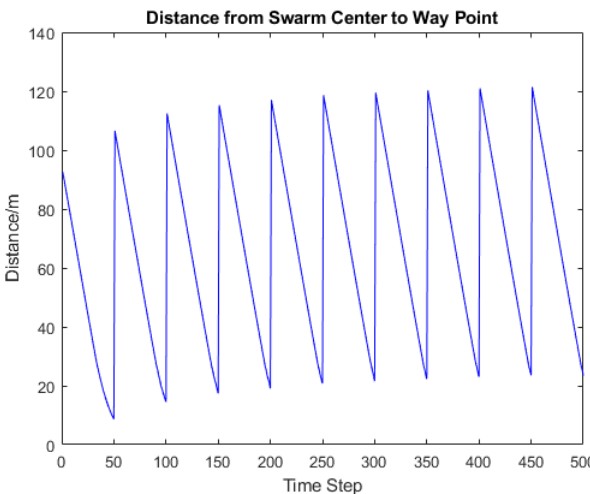

**Figure 18.** Distance between swarm center and waypoint with decentralized control policy.

Figure 19 represents the consistent result with Figure 17. Ten in total communication link disturbance is caused by the waypoint updating and the swarm center transition. The indicator value happens to be stable after the initial transition of the waypoint finishes. For example, it appears at the time step $t = 50, 100, 150$ that the center of the swarm has the lowest distance to the waypoint in Figure 18. Meanwhile, the communication indicator with high highest value starts dropping and approaches the value with stable communication quality to its neighboring agent. Here, a delay effect of such decentralized control seems to take place. Assuming the difference between agent position and waypoint is still large enough, the agent could get affected from both of the waypoints when the first waypoint transition time is not finished. With the increasing historical waypoint on the surveillance map, we wonder if it could cause the disturbance over communication links within the UAV swarm network. The following results would explain it.

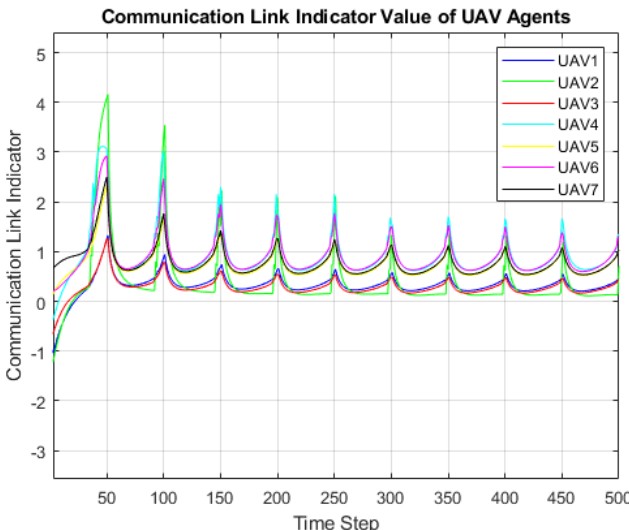

**Figure 19.** Communication indicator variance with decentralized control policy.

As shown in Figure 20, we simulate the swarm track waypoints in a hexagon distribution with fixed transition time to each waypoint and high swarm position control gain, respectively. Our control scheme shows its constraints on realizing its functions in accurate swarm center point tracking with short transition time. In other words, the gain of position updating control for UAV swarm is not enough. The limited coverage area reduces the performance of the swarm in providing continuous surveillance service. The distance

from agent 1 to the others maintains at the range from 20 m to 40 m, which is still an appropriate range for building high throughput rate of data exchange network within the UAV swarm. The swarm center gets further away from the waypoint as time goes on due to not enough gain over control. The swarm center shows the trend of approaching the updated waypoint after time step 400. Compared to the successful waypoint tracking realized with straight-line trajectory yet, the hexagon waypoint tracking with less comprehensive surveillance coverage has a more stable communication link. From the figure, we conclude that the mobility of low altitude UAV swarm is sacrificed by maintaining the stability of swarm communication network.

We did the experiment with increasing the gain for each agent's position control. The performance of UAV reaching designed surveillance waypoint becomes better; however, the high control gain increases the battery consumption in real swarm operation, which is another essential factor affecting successful rate of the UAV swarm's surveillance mission. Here, the power efficiency is sacrificed by the control efficiency. Moreover, the high gain's instability could make the swarm system less controllable on building a stable UAV swarm network. The instability trend of swarm's internal communication link suggests the high QoS variance within the UAV network. As shown in our SISO model and the simulation result above, the higher value of SISO communication indicator suggests the more substantial mutual interference between neighboring antennas. A high package loss rate happens in this situation. However, the oversaturated waypoint tracking gain on control essentially increases the accuracy of waypoint tracking.

Considering each UAV platform's battery consumption, we would also like to avoid the high gain on the agent's position control. Other factors that could also affect the surveillance mission performance include the high altitude UAV's assigned surveillance time and the waypoint distribution. Here, we use the same hexagon distributed waypoints example with optimized transition time and control gain to express this idea. As shown in Figures 21–23, all metrics on the tracking accuracy and agent's communication indicator value are being kept on the relatively low value compared to the results of overfed gain above. The high accurate waypoint tracking could be seen from Figure 22. The distance value is all lower or near 10 m after fast position updating regarding the assigned waypoint. The communication channel quality is controlled by the after adjusting swarm center position. As time goes by, the indicator value stays on the origin after 9th waypoint updates. Figure 21 shows the UAV agent has the distance to the others with appropriate range value.

Figure 24 compares the waypoint tracking accuracy from three algorithms: (1) Centralized Policy (CP); (2) Alternative Tracking with Centralized Policy (ATCP); (3) Decentralized Waypoint Tracking Algorithm. This is based on the different gain value on communication aware control and waypoint tracking control. Decentralized control algorithm could be divided into (a) Decentralized Policy with High Gain on Tracking (DPHGT); (b) Decentralized Policy with Optimal Gains (DPOG); (c) Decentralized Policy without Gain Control (DPWGC). Here, tracking accuracy evaluates the percentage of the reduced distance between swarm center and waypoint position with half of the transition time. Therefore, we have

$$\text{Tracking Accuracy}(\%) \ = \ \frac{d_{\text{median}}}{d_{\text{initial}}} \tag{31}$$

where the $d_{\text{medium}}$ denotes the median of the distance between swarm center and waypoint within one transition period, $d_{\text{initial}}$ denotes the initial value of the distance between swarm center and waypoint in the waypoint transition period. With more accurate waypoint tracking ability, the faster convergence of swarm center and waypoint distance the algorithm would reach. In Figure 24, the DPHGT has the highest tracking accuracy because of the significant gain value on the UAV agent's position control. CP has the lowest tracking accuracy because the UAV agent loses control while following multiple waypoints. DPOG, with the optimally designed gain control on both aspects of communication and waypoint tracking, has a similar tracking accuracy compared to DPHGT. ATCP shows better tracking accuracy after 8th round

of waypoint updating. DPWGC shows less accurate waypoint tracking performance without the adjustment of control gain in following the hexagon distributed waypoints.

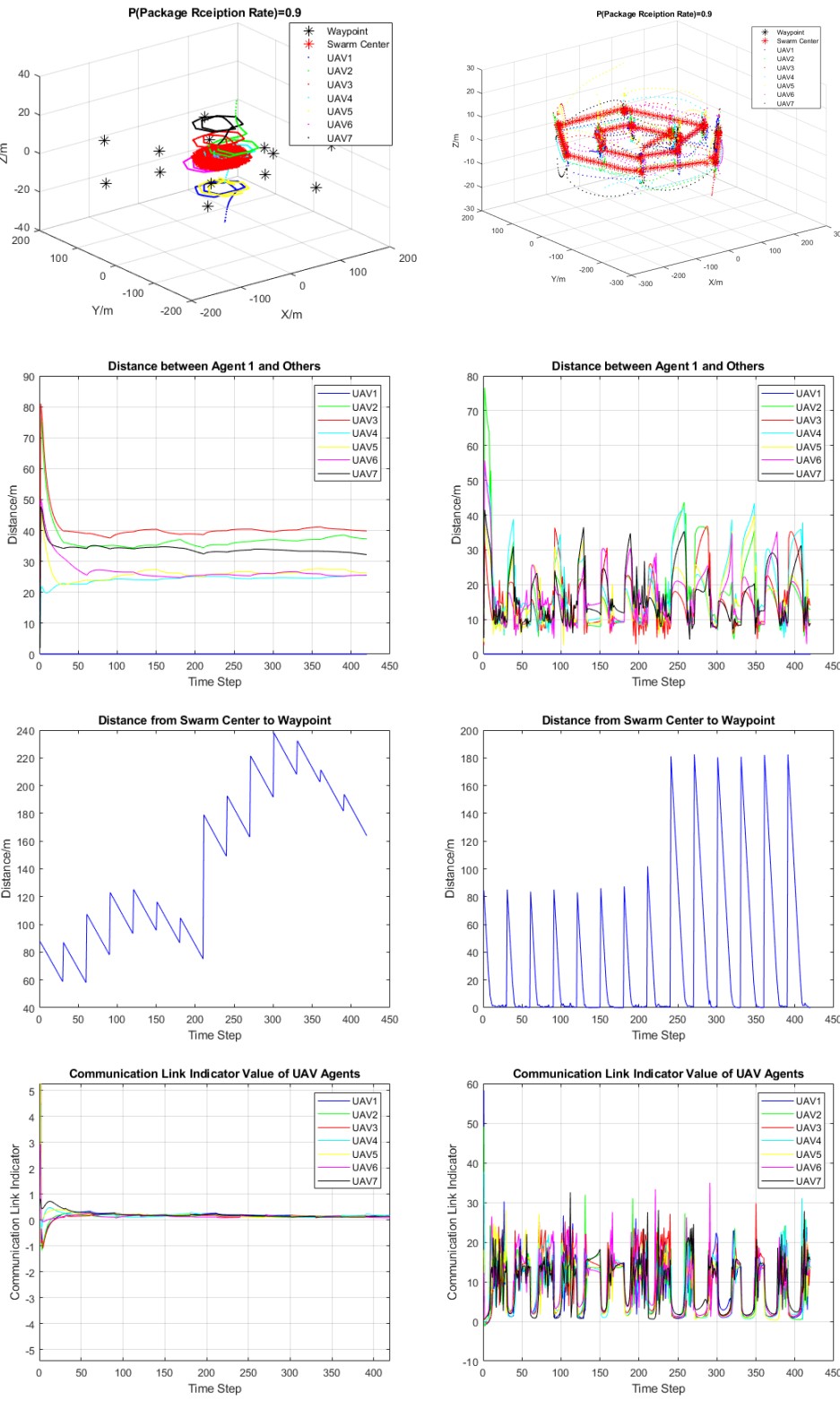

**Figure 20.** Comparison of UAV Swarm Dynamics, Distance between Agents, Distance between Swarm Center and Waypoints, Communication Indicator on not enough transition time (or low position approaching control gain) (left column), high position approaching control gain (right column), respectively.

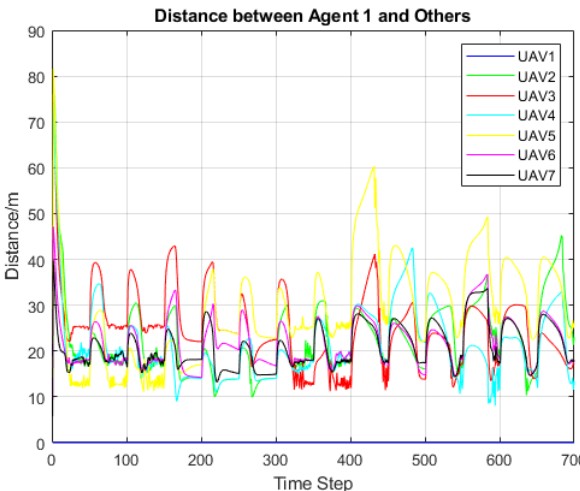

**Figure 21.** Distance between agent 1 and others based on optimized gain and transition time.

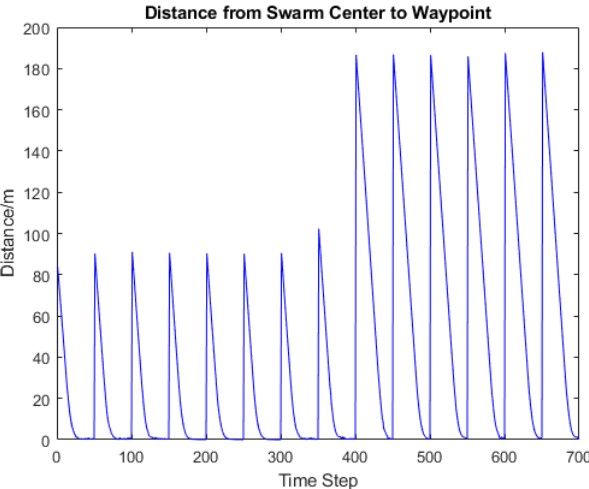

**Figure 22.** Distance between swarm center and waypoint based on optimized gain and transition time.

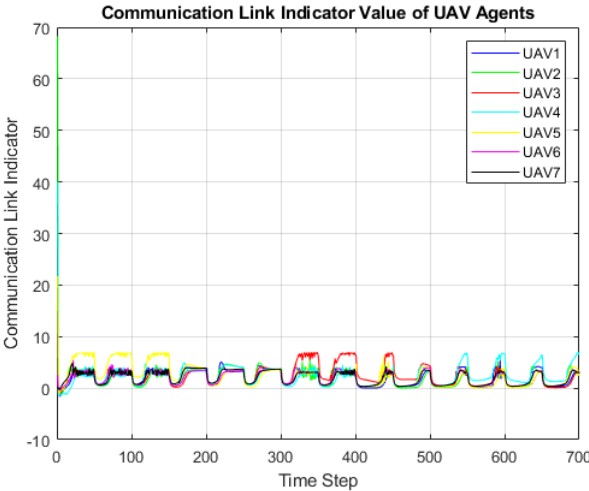

**Figure 23.** Communication indicator variance based on optimized gain and transition time.

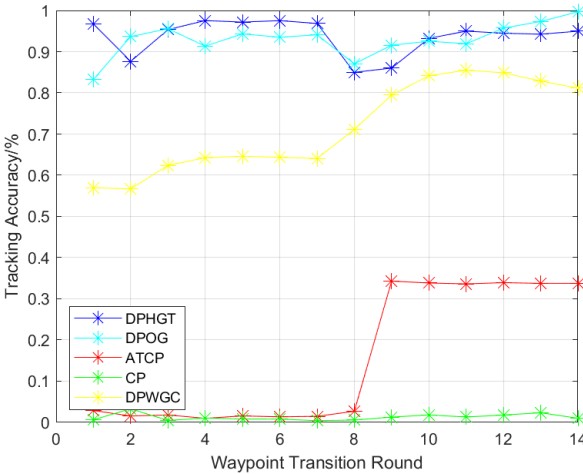

**Figure 24.** Waypoint Tracking Accuracy with Different Algorithms.

The average distance from agent 1 to the others measures the general performance of formation control on each algorithm. A larger average distance value suggests that the algorithm is less performed in forming a surveillance swarm team. It can be represented as

$$D_{avg} = \frac{\sum_{n=1}^{\text{swarm size}} D_{1i}}{swarm\ size} \tag{32}$$

in which, $D_{1i}$ represents the distance between agent one and agent *i*. *swarm size* is chosen as 7 to match our simulation setup. As we can see in Figure 25, the divergence of CP suggests the swarm agent lost control in the process of following surveillance waypoints. DPOG reaches the lowest average distance between UAV agents but also prevents the collision between them.

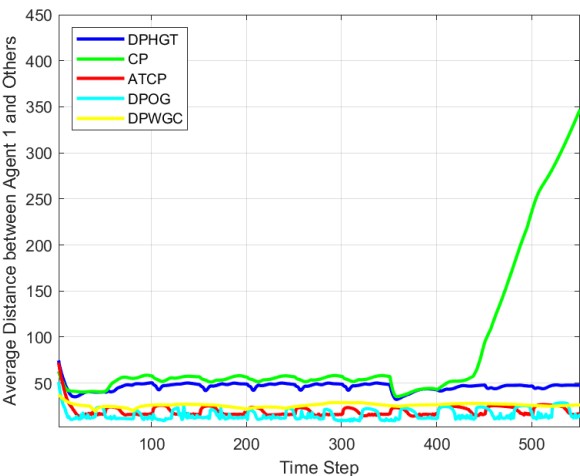

**Figure 25.** Average Distance to UAV Agent 1 with Different Algorithms.

## 4. Conclusions

This paper proposes a communication-aware formation control algorithm to realize high accuracy waypoint tracking with the application in persistent surveillance based on UAV teaming's hierarchical architecture. A framework of such UAV and ground station teaming scheme is proposed in the surveillance strategy. We employed three algorithms to achieve high accuracy waypoint tracking while maintaining communication link quality based on the centralized and decentralized control scheme. The simulation shows that the centralized control scheme reached a fast convergence of maintaining the communication link performance and UAV swarm dynamics stability. In the waypoint tracking, we discussed an alternative centralized control scheme to cope with the divergence of swarm formation. Although the results show that the alternative centralized scheme could

realize a good formation control scheme and network service, the waypoint tracking accuracy of the swarm is reduced, making this scheme hard to implement in a real persistent surveillance mission. Therefore, the decentralized control scheme divides the position updating process from the center to each agent. The simulation results demonstrate the superiority of such decentralized control policy on maintaining the network service quality in terms of communication link stability. Meanwhile, the agents' distance suggests that the UAV swarm's team formation stays in a stable state.

In the simulation result based on the decentralized swarm formation scheme, we studied the trade-off between choosing the waypoint tracking accuracy and maintaining the communication link quality. We discussed the relationship between these two parameters and the battery capacity, algorithm computation efficiency, and thrust of control. With not enough waypoint set on the map for low altitude UAV swarm, the sensing coverage area could not be expanded, which reduces the success rate in persistent surveillance missions. This problem could be solved by adding gain over the position updating control on each agent. However, higher gain on the position updating control cuts down UAV swarm's stability, maintaining an effective communication link with its neighboring agents. Both the balanced gain over two portions on communication quality and tracking accuracy should be considered. In the end, the simulation results showed the optimized gain value, which helps enhance the performance of UAV swarm in persistent surveillance.

In our future research of communication link performance in UAV swarm dynamics, the UAV MIMO antenna's cochannel noise is an inevitable technical challenge to build an effective and efficient UAV swarm network. Cochannel noise is intended to be suppressed with the variance of signal transmission power and arrival angle, especially concerning the application of a millimeter-wave communication system. A more accurate UAV agent dynamic model is desired to simulate its kinetic move within the swarm. Environment interference would be less considered due to the relatively short communication distance between swarm agents. Therefore, millimeter-wave has fewer path loss effects in the process of signal transmission.

Similarly, multipath effects caused by the reflection of wave propagation should be neglected because the UAV swarm operates mostly in the outdoor environment and the existing signal filtering algorithm could eliminate these noises. Through applying persistent surveillance using hierarchical structure UAV teaming with the high altitude UAV platform, the dynamics behavior and communication process and quality evaluation could be more complicated. The simulation of integrating the low altitude UAV swarm and fixed-wing high altitude UAV platform should be done in the future.

**Author Contributions:** Conceptualization, C.X.; methodology, C.X. and Y.J.; software, C.X.; validation, C.X.; formal analysis, C.X.; writing—original draft preparation, C.X.; writing—review and editing, C.X., K.Z., and S.N.; supervision, T.Y. and H.S.; project administration, H.S.; funding acquisition, H.S. All authors have read and agreed to the published version of the manuscript.

**Funding:** This work was supported in part by the National Science Foundation under Grant No. 1956193.

**Institutional Review Board Statement:** Not applicable.

**Informed Consent Statement:** Not applicable.

**Data Availability Statement:** The data and simulation code presented in this study are available on request from the corresponding author. The data are not publicly in this study due to project restrictions.

**Acknowledgments:** We thank Sergey V Drakunov from Embry-Riddle Aeronautical University Department of Engineering Physics for providing insight and expertise in the EP707 course: nonlinear system control, which greatly assisted this research.

**Conflicts of Interest:** The authors declare no conflict of interest.

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
