# Peer review of "Communication Aware UAV Swarm Surveillance Based on Hierarchical Architecture"

_drones, doi:10.3390/drones5020033_

Round 1
Reviewer 1 Report
Dear Authors,
the paper is rather interesting but it needs a deep review. My comments follow:
- several typos and English errors; the written form must be improved. Some sentences are not well understood and the paper is not very fluent to read;
- your contribution is not evident: you talk about "swarm control" but many similar articles are available. On this point some perplexities arise: the use of a potential field (known techinque) appears, but with no explanation;
- A deeper framework is needed: since you propose a swarm magement technique, showing numerical results, you must deepen the introduction and better frame the whole paper;
- The model of the aircraft is extremely simplified. I think that it's more opportune at least a model controlled in acceleration, rate of turn and climb angle which allows the building of a control system already compatible with an aircraft equipped with an autopilot.
- The chapter about results is unwatchable. Some parts of the algorithm are explained but it's not clear why. The graphics are unwatchable and the captions are not clear. A metric is missing to evaluate the performance of the algoritm instead of some figures (12b or 13b or 13c) that are useless.
Author Response
Many thanks for the detailed comments and suggestions for encouraging our work and for giving useful comments for clarifying, improving, and correcting some materials in the paper.
Now, we have carefully revised the paper according to your comments, as explained below.
Point 1: Several typos and English errors; the written form must be improved. Some sentences are not well understood and the paper is not very fluent to read;
Reply: Thanks for pointing this issue out. We have carefully revised the grammar errors and reformatted some of the long sentences hard to understand.
Point 2: your contribution is not evident: you talk about "swarm control" but many similar articles are available. On this point some perplexities arise: the use of a potential field (known techinque) appears, but with no explanation;
Reply: We agree that many swarm control articles are available. The consensus control of UAV swarm is not a novel idea to investigate. However, for the physical layer in the UAV swarm communication, articles have rarely introduced the communication link quality with the swarm’s dynamics. This research can become essential with the popular adoption of MIMO millimeter-wave antenna in the communication of UAV since the millimeter-wave provides high data transmission speed. Surely, the SISO model we adopted neglects the co-channel noise of MIMO and interference caused by DOA, but the communication channel dynamics with kinetic behavior of UAV is what we are interested in in this research. Therefore, we adopted the tools from nonsmooth analysis for designing the controller and stability analysis at the initial status of this research. The potential field is both used in designing the waypoint tracking and communication aware formation. The simulation results show the diversity of introducing potential field techniques in our model.
Point 3: A deeper framework is needed: since you propose a swarm magement technique, showing numerical results, you must deepen the introduction and better frame the whole paper;
Reply: We agree. The numerical results of comparing each algorithm are added at the end of section 3.
Point 4: The model of the aircraft is extremely simplified. I think that it's more opportune at least a model controlled in acceleration, rate of turn and climb angle which allows the building of a control system already compatible with an aircraft equipped with an autopilot.
Reply: Thanks for pointing this out. We agree that the model with control in acceleration would make it be more compatible with the real system. We would like to include it in our next step of research on investigating a more realistic UAV agent model and practical communication system. For this paper, the presenting of communication link dynamics with a simplified UAV model would help us present more complete results of applying UAV swarm in surveillance, especially when considering the communication link quality.
Point 5: The chapter about results is unwatchable. Some parts of the algorithm are explained but it's not clear why. The graphics are unwatchable and the captions are not clear. A metric is missing to evaluate the performance of the algoritm instead of some figures (12b or 13b or 13c) that are useless.
Reply: We agree. Most of the figures have been color-coded and added necessary explanation.

Reviewer 2 Report
Dear authors,
I liked your introduction, background, mathematical model and algorithms. I enjoy the flow of how you explain your method and approach while developing your model. However, there are several details that have to be adjusted in order to have your document considered for publication. In summary, there are missing citations, some Figures are not properly matched or lack a number, your overall figures need color coding, legends that can provide a better visualization for the reader to understand your results. When you compare models or values you need statistical data such as variances or standard deviations that cad validate and verify when you state why one model is more efficient to other. You seem to have run one simulation per each algorithm once and use that as the proof of your model been efficient. For every experimental procedure you need a certain number of trials to verify the replicability of your model. You need maybe running the code for algorithm 1 five times and document its capability to enable the UAVs to reach the target. One run is not enough. Also, why 7 drones, not 6 or 4? How do you base that assumption? Did you have the capability of running a swarm of 7 or there is literature that you refer to get that specific number of drones. In my research lab we work with Parrot Mambo mini drones and we were able to develop a swarm of 5 to 6 drones. And based on communication capabilities and system constraints we could go higher than that. So be specific of how you pick up those values.
You need also to have a control environment and specify assumptions such as that the UAV work in ideal weather conditions (wind will add complexity to the model), that it is assumed that the devices will maintain a safe distance and will not collide with each other. You need to assume that there are not projectiles such as birds in the model and so on. Specify also or give an idea of which type of hardware would you utilize in hands-on test (ex. DJI Mavic, Tello, Parrot Mambo and so on). You need to have in mind how a real application will be implemented. Also, battery consumption have to be assumed since it would specify the time of flight you might have available. Every time you explain that increasing the gain will enhance the capability of your model, more power gets consumed by the device. In that case power efficiency is sacrificed by control efficiency. If many of these variables are outside the scope of your work, specify it. Also, there is not explanation of what software you utilized to model and simulate your control algorithms (MATLAB?). If it was MATLAB you can definitely fixed your graphs, ad different colors, difference line thickness, dotted lines, legends, markers that specify starting points and ending points and so on.
Below is a detail list of issues and other observations in the paper. I stop to a certain point because the problems repeated:
- Line 34 and 44: Use a semicolon (;) not a period to separate the list of items
- Figure 2. Elaborate on the image. Is the black spot fire. How do you obtain that image, is that a concept of operations? Add labels to the arrows and label all the sections of the image or add a color coded legend.
- You developed a good and descriptive introduction.
- What is your research focus in the Hierarchical Architecture? Are you focusing this paper on the High altitude or the Low altitude? You should have described from the beginning that the rest of the paper is focused on the Low altitude to later implement the high altitude fixed-wing UAV.
- Line 1 multi-agent vs. line 136 multi agent. Be consistent with those term. If that word has a hyphen, keep on adding the hyphen.
- Line 142, what is SISO? Remember that this is for a general Drones publication, not an specialized journal in communications so you need to meet or consider a broader audience. If you use acronyms, introduce them. Single-In/Single-Out -> SISO.
- Line 166. Missing reference number
- Line 175 need to introduce the T variable. Every variable shall be introducing when explaining a method or the development of a model.
- Line 230, missing reference numbers
- Equation 10 in line 175 is exactly the same as Equation 29 in line 241. Just refer back to Equation 10 instead of repeating and relabeling an existing one.
- Line 248 and 249, I am interested in the computational efficiency of your algorithms. How long it take to would take a UAV to react based on your algorithm in a real life scenario having in consideration microcontroller delays, interaction with other sensors, the GPS, and so on. We start with an ideal model, but we also need to consider the efficiency of our controls or algorithms.
- What software did you use?
- Figures 4, 5, 6, 7 and so on…. Need to be color coded, need a legend (UAV1, UAV 2…). You need to improve the visualization capability of your document. Also, have in consideration that many researchers like to print the paper and in black and white it will be very hard to read your images or make sense of them.
- There is an inconsistency in the sequence of figures: Figure 8 label repeats twice in two different figures and the consistency is lost.
- Line 303, how you know that between 29 & 50 meters is a good number. How do you validate and verify that assumption?
- When the gain is increase, how it affect the performance of the UAV?
- Line 400, the figure number is missing
- Line 417 to 418, the statement needs to be verified and validated with statistical or probabilistic data.
Author Response
Many thanks for the detailed comments and suggestions for encouraging our work and for giving useful comments for clarifying, improving, and correcting some materials in the paper.
Now, we have carefully revised the paper according to your comments, as explained below.
Point 1: There are missing citations, some Figures are not properly matched or lack a number, your overall figures need color coding, legends that can provide a better visualization for the reader to understand your results.
Reply: We agree. The missed citation has been corrected. Figures lacking a number have been corrected as well. All figures having multi-UAVs have been color-coded, and have legends included.
Changes: Added Figures 4, 5, 6 to explain the model. Figure 7, 9, 10, 11, 12, 13, 14, 15, 16, 17, 18, 19, 20, 21, 22 have all been color-coded and have legends included.
Point 2: When you compare models or values you need statistical data such as variances or standard deviations that cad validate and verify when you state why one model is more efficient to other. You seem to have run one simulation per each algorithm once and use that as the proof of your model been efficient. For every experimental procedure you need a certain number of trials to verify the replicability of your model.
Reply: We agree. At the end of simulation and analysis (section 3), we added the model and algorithm comparison in terms of their tracking accuracy and team formation performance. The newly generated color-coded figures are all based on few times trial running before we generate the final graph.
Point 3: You need maybe running the code for algorithm 1 five times and document its capability to enable the UAVs to reach the target. One run is not enough. Also, why 7 drones, not 6 or 4? How do you base that assumption?
Reply: We agree. We run the algorithm multiple times before adding the figures showing it could reach the target point. The explanation of the swarm number is given in the last paragraph before section 3.1.
Point 4: Did you have the capability of running a swarm of 7 or there is literature that you refer to get that specific number of drones. In my research lab we work with Parrot Mambo mini drones and we were able to develop a swarm of 5 to 6 drones. And based on communication capabilities and system constraints we could go higher than that. So be specific of how you pick up those values.
Reply: The explanation is given in the last paragraph before section 3.1. Reference literature could be found in
Li, Heng, Jun Peng, Weirong Liu, Kai Gao, and Zhiwu Huang. "A novel communication-aware formation control strategy for dynamical multi-agent systems." Journal of the Franklin Institute 352, no. 9 (2015): 3701-3715.
A communication-aware UAV swarm formation control is adopted in their work. The final simulation work shows that 7 agents reach great even distribution of the swarm team.
Point 5: You need also to have a control environment and specify assumptions such as that the UAV work in ideal weather conditions (wind will add complexity to the model), that it is assumed that the devices will maintain a safe distance and will not collide with each other.
You need to assume that there are not projectiles such as birds in the model and so on. Specify also or give an idea of which type of hardware would you utilize in hands-on test (ex. DJI Mavic, Tello, Parrot Mambo and so on).
You need to have in mind how a real application will be implemented. Also, battery consumption have to be assumed since it would specify the time of flight you might have available. Every time you explain that increasing the gain will enhance the capability of your model, more power gets consumed by the device. In that case power efficiency is sacrificed by control efficiency. If many of these variables are outside the scope of your work, specify it.
Reply: We agree. The assumption of our simulation is added at the beginning of section 3.
The increase of power consumption while adding control gain for a position update is added in the analysis of figure 20. We also added the assumption that the UAV battery capacity is large enough to support the flight operation of the UAV.
Point 6: Also, there is not explanation of what software you utilized to model and simulate your control algorithms (MATLAB?).
Reply: Thanks for pointing it out. We explained the simulation environment is in Matlab at the beginning of section 3.
Point 7: Line 34 and 44: Use a semicolon (;) not a period to separate the list of items.
Reply: (;) has been added in the 2nd paragraph of section 1.
Point 8: Figure 2. Elaborate on the image. Is the black spot fire. How do you obtain that image, is that a concept of operations? Add labels to the arrows and label all the sections of the image or add a color coded legend.
Reply: The explanation of the arrow is added in the 7th paragraph and change the figure to present the meaning of the arrow clearer. An explanation of how this wildfire graph is generated is also added in the same paragraph.
Point 9: What is your research focus in the Hierarchical Architecture? Are you focusing this paper on the High altitude or the Low altitude? You should have described from the beginning that the rest of the paper is focused on the Low altitude to later implement the high altitude fixed-wing UAV.
Reply: The focus of research is explained at the end introduction. The discussion of high altitude fixed-wing UAV involved UAV surveillance teaming is presented in the conclusion as our future work.
Point 10: Line 1 multi-agent vs. line 136 multi agent. Be consistent with those term. If that word has a hyphen, keep on adding the hyphen.
Reply: The consistency issue is solved.
Points 11: Line 142, what is SISO? Remember that this is for a general Drones publication, not an specialized journal in communications so you need to meet or consider a broader audience. If you use acronyms, introduce them. Single-In/Single-Out -> SISO.
Reply: SISO is explained in the first paragraph of section 2.
Point 12: Line 175 need to introduce the T variable. Every variable shall be introducing when explaining a method or the development of a model.
Reply: T variable is explained before equation 11.
Point 13: Line 230, missing reference numbers
Reply: Missing reference number is added.
Point: Equation 10 in line 175 is exactly the same as Equation 29 in line 241. Just refer back to Equation 10 instead of repeating and relabeling an existing one.
Reply: Equation 29 is deleted and referred to as equation 12.
Point 14: Line 248 and 249, I am interested in the computational efficiency of your algorithms. How long it take to would take a UAV to react based on your algorithm in a real life scenario having into consideration microcontroller delays, interaction with other sensors, the GPS, and so on. We start with an ideal model, but we also need to consider the efficiency of our controls or algorithms.
Reply: The computation efficiency would not be good on the centralized algorithm since of the delay of control data exchange between swarm agents. Generally, in the centralized policy of swarm, without the information processing service provided by the high-altitude UAV platform (which has a bigger battery capacity), the leader device dealing with most swarm information is not energy efficient. As time goes by, the imbalance of energy capacity for agents would degrade the performance of surveillance missions. The decentralized algorithm is usually having better computation efficiency for each agent since it reduces the control information exchange within the swarm network.
Point 15: What software did you use?
Reply: Matlab
Point 16: Figures 4, 5, 6, 7, and so on…. Need to be color-coded, needs a legend (UAV1, UAV 2…). You need to improve the visualization capability of your document. Also, have into consideration that many researchers like to print the paper and in black and white it will be very hard to read your images or make sense of them.
Reply: All figures have color-coded
Point 17: There is an inconsistency in the sequence of figures: Figure 8 label repeats twice in two different figures and the consistency is lost.
Reply: The inconsistency problem is solved.
Point 18: How you know that between 29 & 50 meters are a good number. How do you validate and verify that assumption?
Reply: An explanation is added in the analysis of figure 15.
Point 19: Line 400, the figure number is missing
Reply: Figure number is added.
Point 20: Line 417 to 418, the statement needs to be verified and validated with statistical or probabilistic data.
Reply: Explanation is added at the end of section 3.

Reviewer 3 Report
This work is focused on developing a decentralized unmanned aerial vehicle (UAV) swarm monitoring approach towards the optimal trajectory design and link quality robustness. The communication model is presented in detail, while the trajectory design schemes are thoroughly documented. Furthermore, the proposed trajectory design schemes are evaluated in terms of distance from the leading UAV, sum-distance among the UAVs, and communication link quality.
The overall organization and structure of the paper are considered good. Nevertheless, there are some issues that have to be addressed:
1) The contributions of this work in the introduction can be enhanced with more details.
2) Section 2 is too short. Therefore, the authors may want to consider merging it with another section.
3) In equation (9), the term ‘q_i’ denotes the i-th UAV position in the 3D space. Then, the summing of positions is irrational. The authors are kindly requested to address this appropriately. For example, is it a sum of distances between the position of the main UAV and the rest?
4) To assist the readers, a paragraph summarizing the evaluation parameters should be included at the beginning of section 4.
5) It seems there are some missing references on page 8, line 230.
6) When referring to distances the corresponding measurement unit should be included. For example, on page 11, lines 288: “A distance over 100 is assumed…” should be “A distance over 100m is assumed…”.
7) How is the communication link quality measured in figures 13, 17, and 18? Is it measured as a pathloss? Also, the appropriate measuring unit (e.g., dB, dBm, etc.) should be added.
8) A figure reference is missing on page 20, line 400.
9) A discussion of future extensions and applications of this work can be included in the conclusion.
10) A revision in terms of the use of the English language should be carried out as there are several grammar errors.
11) More references can be included in this work. The authors may want to consider including the following recent references among others:
[1] Zhou, Xiaobo, Qingqing Wu, Shihao Yan, Feng Shu, and Jun Li. "UAV-enabled secure communications: Joint trajectory and transmit power optimization." IEEE Transactions on Vehicular Technology 68, no. 4 (2019): 4069-4073.
[2] Pliatsios, Dimitrios, Panagiotis Sarigiannidis, Sotirios K. Goudos, and Konstantinos Psannis. "3D Placement of Drone-Mounted Remote Radio Head for Minimum Transmission Power Under Connectivity Constraints." IEEE Access 8 (2020): 200338-200350.
[3] Li, Mushu, Nan Cheng, Jie Gao, Yinlu Wang, Lian Zhao, and Xuemin Shen. "Energy-efficient UAV-assisted mobile edge computing: Resource allocation and trajectory optimization." IEEE Transactions on Vehicular Technology 69, no. 3 (2020): 3424-3438.
[4] Li, Zhiyang, Ming Chen, Cunhua Pan, Nuo Huang, Zhaohui Yang, and Arumugam Nallanathan. "Joint trajectory and communication design for secure UAV networks." IEEE Communications Letters 23, no. 4 (2019): 636-639.
[5] You, Changsheng, and Rui Zhang. "3D trajectory optimization in Rician fading for UAV-enabled data harvesting." IEEE Transactions on Wireless Communications 18, no. 6 (2019): 3192-3207.
[6] Zadeh, Parviz Mohammad, Mohsen Sayadi, and Amirreza Kosari. "An efficient metamodel-based multi-objective multidisciplinary design optimization framework." Applied Soft Computing 74 (2019): 760-782.
[7] Chen, Wuhui, Baichuan Liu, Huawei Huang, Song Guo, and Zibin Zheng. "When UAV swarm meets edge-cloud computing: The QoS perspective." IEEE Network 33, no. 2 (2019): 36-43.
[8] Wang, Xuanxuan, Wei Feng, Yunfei Chen, and Ning Ge. "UAV swarm-enabled aerial CoMP: A physical layer security perspective." IEEE Access 7 (2019): 120901-120916.
[9] Li, Bin, Zesong Fei, Yan Zhang, and Mohsen Guizani. "Secure UAV communication networks over 5G." IEEE Wireless Communications 26, no. 5 (2019): 114-120.
Author Response
Many thanks for the detailed comments and suggestions for encouraging our work and for giving useful comments for clarifying, improving, and correcting some materials in the paper.
Now, we have carefully revised the paper according to your comments, as explained below.
Point 1: English language modification and result analysis improvement
Comment 1: The contributions of this work in the introduction can be enhanced with more details.
Reply: We agree that the original contributions of this work listed are ambiguous and not fully correct. We made modifications to the three contributions we did in this paper.
Changes: The modification is made at the end of the introduction.
Comment 2: To assist the readers, a paragraph summarizing the evaluation parameters should be included at the beginning of section 4.
Reply: We agree. We merged session 2 and session 3 and added the figure to explain the single input and single-output (SISO) antenna model.
Changes: Session 2 is merged with session 3. The SISO model is added on session 2.1.
Comment 3: When referring to distances the corresponding measurement unit should be included. For example, on page 11, lines 288: “A distance over 100 is assumed…” should be “A distance over 100m is assumed…”.
Reply: We agree and the modification has been made.
Comment 5: How is the communication link quality measured in figures 13, 17, and 18? Is it measured as a path loss? Also, the appropriate measuring unit (e.g., dB, dBm, etc.) should be added.
Reply: Thanks for reminding us. The communication link quality is measured by the indicator phi_ij, which fully considers the path loss effect in the antenna far-field and the interference effect in the antenna near-field. So the distance between UAV agents should not be too far or too close. It is a dimensionless variable. The reason why we did not add the communication link indicator in figure 13 is that the simulation results show that the trend of UAV swarm is not able to form a surveillance team with algorithm 1. We think that it is not necessary to present the communication indicator value since the indicator value is not convergent. The communication indicator value for original figure 17 is added.
Comment 6: A discussion of future extensions and applications of this work can be included in the conclusion.
Reply: We agree and the extension has been added at the end of the conclusion.
Comment 7: A revision in terms of the use of the English language should be carried out as there are several grammar errors.
Reply: Thanks for pointing it out. We fixed most of our grammar errors in the papers.
Point 2: Format and Structure of the paper
Comment 1: Section 2 is too short. Therefore, the authors may want to consider merging it with another section.
Reply: We agree. Session 2 is merged with session 3.
Comment 2: It seems there are some missing references on page 8, line 230.
Reply: Thanks. The missing reference has been added.
Comment 3: A figure reference is missing on page 20, line 400.
Reply: The missed figure reference is added.
Point 3: Equation Irrational
Comment 1: In equation (9), the term ‘q_i’ denotes the i-th UAV position in the 3D space. Then, the summing of positions is irrational. The authors are kindly requested to address this appropriately. For example, is it a sum of distances between the position of the main UAV and the rest?
Reply: We agree that the sum of "q_i" is irrational and it should be the sum of position value on "x_i, y_i,z_i"
Changes: The changes have been made in session 2.2 and algorithms 1, 2, 3.
Points 4: Add of reference in the literature review
Comment 1: More references can be included in this work. The authors may want to consider including the following recent references among others:
Reply: Thanks for pointing out we are short of enough reference in the paper. We renewed the reference list and added it to the introduction, problem formulation, and simulation.

Round 2
Reviewer 1 Report
Dear Authors, the paper was improved in respecto to the first version, but now it is too long: the reader is very tired at the end of the article; some parts are verbose, the concepts are repeated several times. I think that the paper needs to be summarized. On the other hand, the limitation of the absence of experimental results persists: the simulation is certainly fundamental but it is necessary that there is a minimum experimental verification: the risk is to continue with numerical simulations that will provide non-applicable results.
Author Response
Many thanks for the comments and suggestions for encouraging our revised manuscript.
We have carefully considered your suggestion and revised the paper according to your comments, as explained below.
Point 1: The paper length is too long. Some concepts are repeated several times. The paper needs to be summarized.
Response: Thanks, we agree on the paper length at this point and some of the simulation results are explained with similar concepts. We made the changes as follows.
- In section 3.3, we combined the two simulation results together on 1) low position approaching control gain; 2) high position approaching control gain, respectively. We rewrote the analysis of the result to prevent the repeats of the concepts
- We deleted the part of the wildfire spread simulation result in the introduction since we found out it might not be necessary for forming a descriptive introduction.
Point 2: The limitation of the absence of experimental results persists: the simulation is certainly fundamental but it is necessary that there is a minimum experimental verification: the risk is to continue with numerical simulations that will provide non-applicable results.
Response: Thanks for pointing this out. We had a relatively short time to conduct such an experiment on a UAV swarm platform. The system configuration of existed UAV platform could hardly match our assumption on the MIMO interference aspect. We would like to extend our work in the next step of using a real UAV platform in collecting the real communication performance data, include the MIMO antenna performance, interference suppressing algorithms with the UAV swarm dynamics, etc.

Reviewer 2 Report
You make the proper changes. I see a better organization on your work and the inclusion of a Problem Formulation section helps. I hope you continue with your efforts and I am looking forward to your progress.
Author Response
Thank you for your kind words! This is very encouraging for us! We would like to extend our work in the next step of using a physical UAV swarm platform in collecting the communication performance data, include the MIMO antenna performance, interference suppressing algorithms with the UAV swarm dynamics.
Thank you again for your detailed review comments!
